# Exploring the Intersection of ADHD and Music: A Systematic Review

**DOI:** 10.3390/bs15010065

**Published:** 2025-01-13

**Authors:** Phoebe Saville, Caitlin Kinney, Annie Heiderscheit, Hubertus Himmerich

**Affiliations:** 1Centre for Research in Eating and Weight Disorders (CREW), Department of Psychological Medicine, Institute of Psychiatry, Psychology & Neuroscience (IoPPN), King’s College London, London SE5 8AB, UK; phoebedsaville@gmail.com (P.S.); caitlinkinney11@gmail.com (C.K.); 2Paediatric Psychology Team, Dingley Child Development Centre, Berkshire Healthcare NHS Foundation Trust, Reading RG6 6BZ, UK; 3Cambridge Institute for Music Therapy Research (CIMTR), Anglia Ruskin University, Cambridge CB1 1PT, UK; annie.heiderscheit@aru.ac.uk; 4South London and Maudsley NHS Foundation Trust, London SE5 8AZ, UK; 5Bundeswehr Center for Military Mental Health, Military Hospital Berlin, 10115 Berlin, Germany

**Keywords:** ADHD, ADD, music processing, music perception, music therapy, music-based intervention, audio stimulation

## Abstract

Attention Deficit Hyperactivity Disorder (ADHD) is a highly prevalent neurodevelopmental disorder, affecting both children and adults, which often leads to significant difficulties with attention, impulsivity, and working memory. These challenges can impact various cognitive and perceptual domains, including music perception and performance. Despite these difficulties, individuals with ADHD frequently engage with music, and previous research has shown that music listening can serve as a means of increasing stimulation and self-regulation. Moreover, music therapy has been explored as a potential treatment option for individuals with ADHD. As there is a lack of integrative reviews on the interaction between ADHD and music, the present review aimed to fill the gap in research. Following PRISMA guidelines, a comprehensive literature search was conducted across PsychInfo (Ovid), PubMed, and Web of Science. A narrative synthesis was conducted on 20 eligible studies published between 1981 and 2023, involving 1170 participants, of whom 830 had ADHD or ADD. The review identified three main areas of research: (1) music performance and processing in individuals with ADHD, (2) the use of music listening as a source of stimulation for those with ADHD, and (3) music-based interventions aimed at mitigating ADHD symptoms. The analysis revealed that individuals with ADHD often experience unique challenges in musical tasks, particularly those related to timing, rhythm, and complex auditory stimuli perception, though these deficits did not extend to rhythmic improvisation and musical expression. Most studies indicated that music listening positively affects various domains for individuals with ADHD. Furthermore, most studies of music therapy found that it can generate significant benefits for individuals with ADHD. The strength of these findings, however, was limited by inconsistencies among the studies, such as variations in ADHD diagnosis, comorbidities, medication use, and gender. Despite these limitations, this review provides a valuable foundation for future research on the interaction between ADHD and music.

## 1. Introduction

### 1.1. Attention Deficit Hyperactivity Disorder (ADHD)

Attention Deficit Hyperactivity Disorder (ADHD) is defined by the Diagnostic and Statistical Manual of Mental Disorders, 5th edition (DSM-5), and the International Classification of Diseases, 11th edition (ICD-11), as a persistent pattern of inattention and/or hyperactivity-impulsivity that significantly interferes with an individual’s functioning or development ([5]; [83]). ADHD is classified into three distinct subtypes: the predominantly inattentive type, which comprises 20–30% of cases; the predominantly hyperactive/impulsive type, which comprises 15% of cases; and the combined type, which comprises 50–75% of cases ([48]). As a neurodevelopmental disorder, ADHD is characterised by the presence of significant symptoms of inattention and/or hyperactivity-impulsivity before the age of twelve ([5]; [83]). The presentation of these symptoms tends to vary characteristically across developmental stages, with hyperactivity being more pronounced in preschool years, and inattention becoming more prominent in school-aged children and persisting into adulthood ([23]).

A diagnosis of ADHD requires the presence of at least six symptoms of inattention and/or six symptoms of hyperactivity/impulsivity over a period of six months, with symptoms causing substantial disruption to academic, occupational, or social functioning ([5]). The inattentive subtype is characterised by short attention spans and distractibility, while the hyperactive-impulsive subtype is classified by excessive motor activity and impulsive behaviours ([83]). These behaviours must be inconsistent with the individual’s developmental level and not attributable to other mental health conditions ([5]). Additionally, ADHD is associated with a broad range of adverse outcomes, including impaired social, educational, and occupational functioning, increased involvement with the criminal justice system, risky sexual behaviours, substance and alcohol abuse, and comorbid mental health disorders ([33]). Indeed, at least 75% of children and adolescents with ADHD develop a co-occurring mental health disorder, which complicates diagnosis and treatment and can worsen the overall prognosis ([8]).

ADHD affects approximately 5% of children globally, making it the most prevalent behavioural health condition in this population ([25]). In the United States, ADHD prevalence in children is estimated at 8.7%, raising questions about whether this reflects improved recognition or potential overdiagnosis ([65]). Although once considered a disorder confined to childhood and adolescence, ADHD is now understood to often persist into adulthood, with a global prevalence estimated at 3–4% among adults ([51]; [49]). Significantly, ADHD disproportionately affects males, who are two to five times more likely to be diagnosed than females ([21]; [47]).

The National Institute for Health and Care Excellence (NICE) stipulates an age-specific multimodal treatment approach for ADHD that combines pharmacological and psychosocial interventions ([48]). For children under five, parent-training programmes are advised, as pharmacological treatments are not recommended due to limited research on stimulant use in this age group ([76]; [81]). For children aged five and older, parent-training programmes remain foundational; however, methylphenidate, an agent that enhances norepinephrine and dopamine transmission, is also recommended as a first-line pharmacological treatment ([48]; [13]). Indeed, 70% of children with a current diagnosis of ADHD receive pharmacotherapy ([77]). Cognitive Behaviour Therapy (CBT) is recommended when medication is insufficient, to target cognitive distortions and improve social skills, emotional regulation, and executive functioning ([30]). For adults, similarly, medications like methylphenidate or lisdexamfetamine are recommended, with CBT as a supplement when necessary ([48]; [29]). While stimulant medications are effective, their long-term use is limited by side effects, social stigma, and adherence challenges, with up to half of patients discontinuing within three years ([36]; [88]). Psychosocial interventions, including CBT, are effective alternatives or adjunctive treatments, especially for younger children or when medication is unsuitable ([48]; [21]). However, their long-term success remains limited, indicating a need for more comprehensive, minimally invasive strategies ([22]).

### 1.2. The Effect of ADHD on Music Processing and Performance

Recent neuropsychological models suggest that deficits in ADHD can be traced to disruptions in three neural pathways: the dorsal frontostriatal pathway, associated with cognitive control; the ventral frontostriatal pathway, linked to reward processing; and the frontocerebellar pathway, which influences temporal processing ([71]). These neural differences contribute to phenotypic symptoms such as inattention, impulsivity, and impaired working memory, which can significantly affect music perception and performance ([41]).

Individuals with ADHD can experience difficulties in processing temporal information, a critical skill in music comprehension ([68]). There is strong evidence for an association between these timing deficits and the core symptoms of impulsivity and inattention, suggesting that timing problems are key to the clinical behavioural profile of ADHD ([69]; [75]). Moreover, deficits in sustained attention can negatively impact tasks such as music practice and performance, where maintaining consistent rhythm and timing is essential ([68]). Working memory impairments can further hinder musical learning by making it difficult for individuals to retain and manipulate musical information, affecting their ability to learn new pieces, refine them through practice, and perform consistently ([34]). Additionally, slower auditory processing speeds in those with ADHD can impair the perception of musical elements, complicating tasks such as sight-reading and ensemble playing ([10]).

However, these challenges may indicate the potential for music interventions to help individuals with ADHD. Musical training, particularly through learning to play an instrument, can enhance timing skills and promote auditory cortex development, offering a promising avenue for mitigating the deficits associated with ADHD ([6]). This suggests that targeted music interventions could support individuals with ADHD in overcoming challenges in music perception and performance.

### 1.3. The Effect of Listening to Music on Individuals with ADHD

The relationship between ADHD and music listening is shaped by theories of arousal regulation, such as the optimal stimulation theory and the Moderate Brain Arousal (MBA) model ([86]; [67]). These theories propose that individuals with ADHD often experience a baseline level of under-arousal, prompting them to seek external stimuli to achieve optimal arousal levels necessary for effective functioning ([86]; [67]). Music, as a structured auditory stimulus, provides a source of external stimulation that could help regulate arousal and improve cognitive performance by modulating dopamine levels in the brain ([67]; [44]).

Research demonstrates that music listening can activate multiple brain regions, including those involved in sensory processing, motor control, and the brain’s reward circuitry ([44]; [84]). Neuroimaging studies suggest that music stimulates the dopaminergic pathways, including the nucleus accumbens, which is crucial for motivation and reward processing ([11]). This stimulation may be particularly beneficial for individuals with ADHD who often exhibit deficiencies in dopamine receptor function due to specific disruptions in the ventral frontostriatal pathway ([71]; [67]; [12]).

While external stimuli might distract children with ADHD and negatively impact academic performance, research increasingly shows that carefully selected music can enhance cognitive functioning ([26]; [56]). In fact, evidence suggests that auditory stimuli, such as music, can help individuals with ADHD filter out distractions, aiding in sustaining attention, and improving memory ([70]). Furthermore, moderate levels of background music have been found to improve cognitive performance in children with ADHD, suggesting potential therapeutic benefits ([70]).

Overall, this growing body of research underscores the potential value of music in addressing the sensory and cognitive needs of individuals with ADHD. Music in the form of therapy, in particular, may offer an engaging and motivating treatment option, especially for adolescents who often struggle with treatment adherence ([15]).

### 1.4. The Potential of Music Therapy as an ADHD Treatment

Music is a fundamental aspect of human culture, present across all societies ([14]). Beyond a communicative role, music’s medicinal value has been recognised since 4000 BC, due to its numerous psychological and physiological health benefits ([39]).

Music therapy is a reflexive process in which a qualified music therapist designs, structures, and facilitates musical experiences to address individualised therapeutic goals, using receptive, recreative, composition, and improvisation methods ([31]). These musical experiences may include actively engaging in making music, such as instrumental improvisation, songwriting and singing. Music therapy sessions can also include passively engaging in music listening, which engages diverse brain networks across the cortex, subcortex, and cerebellum ([46]; [17]). The primary aim of music therapy is often to address non-musical needs or skills, rather than develop one’s musical abilities ([63]). Music therapy has been understood to leverage dopamine release, thereby creating a robust learning environment for non-musical tasks and behaviours ([63]).

Music therapy could provide an effective complement to ADHD medications and psychotherapies, as it has been shown to reduce ADHD symptoms and improve various functional domains ([45]). Active engagement in music during music therapy is suggested to enhance social skills and decrease aggression and impulsivity, while passive engagement with music in music therapy may improve academic abilities and attention and reduce disruptive behaviours ([45]). Its therapeutic efficacy is hypothesised to stem, in part, from music’s ability to enhance neuroplasticity by elevating dopamine levels, thereby addressing the dopamine deficits frequently observed in individuals with ADHD ([78]).

Compared to CBT, music therapy offers a flexible treatment option for ADHD that requires less sustained attention, potentially reducing dropout rates, a significant consideration given that up to 80% of adults with ADHD do not complete their first year of treatment ([64]). Indeed, cognitive, behavioural, and logistical challenges, as well as negative treatment attitudes, often hinder participation in structured therapies like CBT, contributing to the high dropout rates ([66]). Such challenges may not be as prevalent in music-based interventions, which in healthcare are considered to be acceptable for patients of all ages ([62]).

The individualised and reflexive nature of music therapy could potentially make it both a valuable standalone treatment and a complement to other recommended therapies. Given that standard multimodal treatments frequently do not lead to full recovery in individuals with ADHD ([30]), incorporating music therapy may enhance overall treatment outcomes.

### 1.5. Aims and Objectives

One systematic review specifically on music and ADHD has been conducted by [45] ([45]). The review employed a relatively narrow search strategy, which resulted in the identification of a limited number of articles from scientific databases like PubMed. To broaden their scope, the researchers included “grey literature” from sources such as Google Scholar and WorldCat, which may not adhere to the rigorous standards of scientific publications ([18]). Moreover, their review primarily focused on the application of music for use in video games. While their findings offered promising insights into the relationship between music and ADHD, they should be interpreted with caution due to these methodological limitations, highlighting the need for further study.

Therefore, to the best of our knowledge, this is the first systematic review to thoroughly examine the effects of ADHD on music comprehension and performance, as well as how various forms of music engagement, whether as a source of auditory stimulation or as a therapeutic intervention, affect individuals with ADHD.

## 2. Materials and Methods

### 2.1. Search Strategy

This systematic review followed the Preferred Reporting Items for Systematic Reviews and Meta-analyses (PRISMA) standards ([54]) using the PRISMA 2020 checklist (see Appendix A). Following initial scoping searches, three reliable scientific databases, PubMed, Web of Science, and PsychInfo (Ovid), were consulted to select relevant papers, following initial scoping searches.

The general search terms were as follows:Attention Deficit Hyperactivity Disorder, ADHD;Attention Deficit Disorder, ADD;Music, music intervention, music therapy.

However, we made adaptations for these general search terms according to the requirements of each electronic database as explained below. Given the limited literature published on music and ADHD, the search terms were deliberately open to capture as many studies as possible. The Boolean operator “AND” was used in each search to combine the search phrases. Each synonymous search term was also connected by the Boolean operator “OR”. For the Web of Science, we also added the truncation symbol (*) after the root Therap *, so multiple variants could be searched for all at once such as ‘Therapies’, ‘Therapist’ and ‘Therapy’.

The following search terms were used; Web of Science: (ALL = (ADHD OR Attention Deficit Hyperactivity Disorder OR Attention Deficit Disorder OR Attention Deficit Disorder with Hyperactivity OR Attention Deficit)) AND (ALL = (Music OR Music Therapy OR Music Therap* OR Music Intervention OR Music-based Intervention)); Pubmed: (ADHD OR “attention deficit hyperactivity disorder” OR “attention deficit” OR “attention deficit disorder” OR “ADD” OR “hyperactivity disorder” OR “Attention Deficit Disorder with Hyperactivity”[Mesh]) AND (music OR “music therapy” OR “music therap*” OR “music intervention” OR “music-based intervention” OR “music therapy”[Mesh]); PsychInfo (Ovid): (attention deficit disorder with hyperactivity.mp. or exp Attention Deficit Disorder with Hyperactivity/or exp Attention Deficit Disorder/or attention deficit.mp. or hyperactivity disorder.mp. or ADHD.mp.) and (exp Music/or music.mp. or music therapy.m.p. or exp Music Therapy/or music intervention.mp. or music-based intervention.mp.).

### 2.2. Inclusion and Exclusion Criteria

The search strategy aimed to identify and review research evidence concerning music and ADHD. Based on this activity, the following prespecified eligibility criteria for including and rejecting research studies in the literature review were established:Inclusion Criteria
-Studies were original articles;-Studies were published in English or German or had a full English-language translation available;-Study participants were humans;-Study participants were diagnosed with ADHD, ADD, or Hyperactivity;-Studies included music performance and processing; music as a source of stimulation; or as a therapeutic intervention.
Exclusion Criteria
-Case studies, conference articles, systematic reviews, or meta-analyses;-Studies that did not report results or clinical outcomes;-Studies where music was not linked to ADHD;-Studies where the results for participants with ADHD were not specifically reported;-Animal studies.


### 2.3. Screening

A two-stage screening process was employed, involving a preliminary title and abstract screening, followed by a comprehensive screening of the full text. Unfortunately, the full text of 11 studies could not be accessed despite extensive efforts to obtain them, including contacting authors when possible and searching multiple databases. As a result, these studies were excluded from the final analysis.

### 2.4. Data Extraction

In the first stage of the screening process, duplicates were removed using EndNote. Then, all abstracts of potential studies were imported into Ryaan, where any remaining duplicates were eliminated. Two independent reviewers (PS, CK) examined the titles and abstracts to confirm they met the eligibility criteria, ensuring inter-rater reliability.

In stage two, relevant information was manually extracted from the studies. Additionally, the methodological quality of the studies was evaluated using the Joanna Briggs Institute (JBI) Critical Appraisal Tools for randomised controlled trials and quasi-experimental studies ([38]); see Appendix A. Only studies that met at least six of the quality assessment criteria were included in the review.

### 2.5. Type of Data Extracted

Publication characteristics: Year of publication, author, and country the study was conducted in.Sample characteristics: Description of the sample such as the sample size, mean age, age range, diagnostic tool, and medication use.Intervention characteristics: Description of the music intervention, type of music, experimental conditions, description of outcome measures, and the study design used.Outcome characteristics: Primary outcomes, such as the incidence or severity of the core symptoms of inattention, impulsivity, and hyperactivity, and the adverse outcome of disruptive behaviour ([5]; [83]). Secondary outcomes, such as academic performance, family and social outcomes, quality of life, or comorbid disorders.

### 2.6. Ethical Considerations

Ethical approval was not required because systematic reviews do not produce original literature; rather, they merely synthesise and evaluate data from publicly available studies.

### 2.7. Analysis

To summarise and compare the included studies, the guidelines for narrative synthesis proposed by Popay et al. were followed ([58]). The studies were organised into tables and grouped by theme, which facilitated the collection of articles with similar research questions or objectives. These studies were then compared and discussed within their respective research areas.

## 3. Results

### 3.1. Search Results

Using the above search methods, a total of 700 papers were identified, which was reduced to 567 papers after duplicate articles were removed. Figure 1 depicts the PRISMA Flow Diagram of the selection process. Table 1, Table 2 and Table 3 depict all included studies (*n* = 21), which encompassed 1250 participants, 881 of whom had ADHD or ADD.

### 3.2. Study Characteristics

This review included data from 20 studies published between the years of 1981–2023. Included studies were published in Brazil (*n* = 2), China (*n* = 3), France (*n* = 1), Germany (*n* = 1), Iran (*n* = 1), Israel (*n* = 1), Latvia (*n* = 2), New Zealand (*n* = 2), Republic of Korea (*n* = 1), South Africa (*n* = 1), Taiwan (*n* = 1) and USA (*n* = 4), spanning 6 continents. Five of the studies were randomised controlled trials (RCTs), in which individuals with ADHD were randomly assigned to either the control or observation group. The remaining 15 studies used either a between-subjects or within-subjects experimental design, using a group of Typically Developing (TD) individuals as a control or no control group.

### 3.3. Study Results

#### 3.3.1. Music Processing and Performance

Table 1 displays a summary of the studies on music processing and performance. Two studies examined the timing deficits in individuals with ADHD regarding music processing. [16] ([16]) used music to investigate time processing in children with ADHD and age-matched TD controls. Children with ADHD performed worse in estimating short time intervals compared to the control group, suggesting that ADHD may impair the ability to process brief time intervals accurately. When comparing musical excerpts of the same duration, children with ADHD perceived tracks as longer when the musical notes had longer durations. This indicates that children with ADHD might process time differently from TD children regarding more complex auditory stimuli like music. Similarly, [59] ([59]) sought to investigate the perceptual and sensorimotor timing skills of children with ADHD and TD controls. Children with ADHD showed general difficulties in perceiving single durations and in tracking the beat, relative to the matched controls. Together, these results highlight timing deficits shown in tasks with music, a function typically associated with the basal ganglia.

[28] ([28]) conducted a study to assess the behavioural and neuropsychological correlates of musical performance in adolescents with ADHD, ADD, and Dyslexia. The results revealed a significant difference in musical performance across groups. In general, the TD control and ADD and ADHD groups scored higher than the dyslexic participants in almost all measures of musical performance, except for rhythmic and pitch memorisation tasks, in which all groups scored similarly. Even though rhythm-related and musical performance deficits have been reported in individuals with ADHD, in this study, adolescents with ADHD and ADD scored similarly to controls in rhythmic improvisation and musical expression.

[27] ([27]) later looked at individual differences in complex music perception in adults with ADHD. The ADHD group was compared to two groups of TD controls; those who were musically naïve and those musically educated. The results showed that adults with ADHD had deficits in the perception of complex music compared to both TD groups. Interestingly, Short Term Memory (STM) capacity was not impaired in young adults with ADHD, suggesting that memory deficits may not persist into adulthood. In addition, the ADHD group overestimated their performance competence compared to both control groups. The findings suggest that individuals diagnosed with ADHD would benefit from special training that not only focuses on improving performance in perceptual musical skills but also involves metacognitive training to develop realistic self-assessment skills.

In summary, ADHD may cause difficulties in tasks involving timing, rhythm, and beat synchronisation ([16]; [59]) and poor performance in musical perception tasks ([59]; [27]). Despite these deficits, ADHD may not limit ability in creative domains such as musical improvisation and expression ([28]).

#### 3.3.2. Music Listening

Table 2 displays a summary of the studies utilising music listening. Two studies investigated the impact of music on arithmetic performance in children with ADHD. [1] ([1]) found that boys with ADHD demonstrated improved arithmetic performance when listening to music compared to silence or speech. In contrast, the TD control group performed similarly regardless of audio stimulation, supporting the optimal stimulation theory as an explanation for ADHD. Similarly, [26] ([26]) found that listening to music improved accuracy in mathematical performance in children with ADHD. However, they found the same improvement in their group of TD controls. This contrasts with the findings of [1] ([1]), thus, questioning the credibility of the optimal stimulation theory as an explanation for ADHD, proposing that it might apply to all children, not just those with ADHD ([26]; [1]).

Similarly, two studies explored the impact of music on reading comprehension in individuals with ADHD. [43] ([43]) examined the effects of different types of music on reading comprehension, specifically in preadolescents with ADHD and age-matched TD controls. The results showed that all music conditions improved reading comprehension in the ADHD group, while performance deteriorated in the control group. Notably, calm music, with or without lyrics, was most effective in improving performance for the ADHD group, whereas rhythmic music had a lesser impact. Additionally, heart rate variability (HRV) was studied as an indicator of autonomic nervous system activity, finding that listening to music reduced HRV changes in the ADHD group, suggesting a regulatory effect on physiological stress responses. Complementary to [1]’s ([1]) conclusions, this suggests that music aids autonomous regulation in children with ADHD, aiding focus, while it may distract TD children ([1]). Furthermore, [20] ([20]) conducted a study examining the impact of music on reading tasks of varying difficulty levels using a large sample of children with ADHD. The findings showed that music had no significant effect on easier reading tasks; however, music had a negative impact on more difficult reading tasks, suggesting that music might distract students with ADHD when the cognitive load of a task is high. These studies together suggest that while music can be beneficial to the reading comprehension ability of those with ADHD in certain contexts, it may hinder performance when the tasks or musicality require substantial cognitive resources.

Additionally, two studies investigated the impact of music on the motor activity levels of children with ADHD traits. [82] ([82]) conducted a study with a sample of 13 hyperactive boys to investigate the effects of listening to a short musical stimulus. Increased motor activity was observed in the children during the audio stimulation, which was potentially due to the length of the class period rather than the music intervention itself. The study concluded that as activity levels increased throughout the class period regardless of the music intervention, music’s impact on activity levels may be negatively influenced by external factors. Five years later, [19] ([19]) conducted a study, examining the impact of instrumental rock music on hyperactive boys diagnosed with hyperkinetic disorder and/or ADD. Conversely, the results showed a significant reduction in motor activities when the children listened to the music, emphasising that [82]’s ([82]) contrasting results may stem from methodological limitations.

More recently, [7] ([7]) conducted a study to explore the effects of auditory stimulation on upright balance performance in children with ADHD. The intervention compared the effects of soothing music, silence, and white noise on children with ADHD and TD controls. The study found that both music and white noise enhanced upright balance in children with ADHD, suggesting that auditory stimulation, such as music, could have potential benefits for motor tasks and physical coordination.

[56] ([56]) investigated the impact of contemporary rock or rap music on behaviour in boys with ADHD. The study found mixed results, with some children showing improvement in behavioural intervention performance, while others showed no change. The findings indicate individual differences in response to music as a distractor. This suggests that music may help some children with ADHD more than a silent environment, but the response to music or different music genres varies among individuals.

Finally, [90] ([90]) conducted an RCT to investigate the effect of Mozart’s music on the mood of adults with ADHD compared with TD controls. Participants were randomly assigned to the music intervention group, which involved listening to Mozart’s music or the control group, which involved silence. Listening to Mozart decreased negative moods in both those with ADHD and those without. Additionally, music listening helped reduce disruptive behaviours and improve focus in some individuals with ADHD, supporting the idea that music listening can aid autonomous regulation. Furthermore, silence negatively impacted people with ADHD, increasing arousal and decreasing positive mood. Therefore, the study suggests that music listening can help manage both mood and behavioural regulation in ADHD.

In summary, music listening appears to improve mathematical performance ([43]; [1]), simple reading task performance ([43]), and upright balance performance ([7]), whilst reducing hyperactivity symptoms ([19]) and improving mood in people with ADHD ([90]). However, music listening may distract people with ADHD if a cognitive task is very challenging ([20]) and increase motor activity in the presence of adverse external factors ([82]). Overall, although there is a plethora of positive outcomes regarding the impact of music, it must be noted that the response to music varies among individuals ([56]).

#### 3.3.3. Therapeutic Effects of Music and Music Therapy

Table 3 summarises the studies on the therapeutic effects of music and music therapy. Two studies specifically investigated the potential benefits of an intervention combining music with physical activity for individuals with ADHD ([40]; [42]). [40] ([40]) investigated the effects of a music and movement intervention on children with ADHD. The study found significant improvements in the participants’ quality of life post-intervention, indicating enhanced well-being. Additionally, there was a significant reduction in reaction times, suggesting improved attention capabilities. Finally, EEG measurements revealed increased alpha power and decreased delta power, which are associated with enhanced brain functioning. Based on these outcomes, the study concluded that music and movement interventions could effectively improve the quality of life and attention in children with ADHD, supporting their use as a therapeutic option. [42] ([42]) conducted a study with a sample of kindergarten children diagnosed with comorbid ADHD and Oppositional Defiant Disorder (ODD) using an intervention that consisted of a combination of yoga and music. The results demonstrated that the combined intervention produced a significant reduction in symptoms of inattention and hyperactivity/impulsivity. Moreover, there was a decrease in ODD scores post-intervention, reflecting enhanced behavioural regulation. These findings demonstrate that the holistic approach of the combined yoga and music intervention can be an effective strategy for managing these conditions. Together, these studies highlight the benefits of integrating music with physical activities, such as movement and yoga, for improving attention, behaviour, and well-being in children with ADHD.

A study by [55] ([55]) conducted an RCT to examine the effects of music therapy on children with comorbid ADHD and depression. Participants were randomly assigned to either an ADHD control group receiving standard care or an ADHD music therapy group that participated in bi-weekly music therapy sessions, which included both active and passive activities. The music therapy group showed increased serotonin levels, indicating improved mood regulation and decreased cortisol levels, blood pressure, and heart rate, indicating reduced stress and physiological arousal. Furthermore, there were significant reductions in depression and daily stress levels, reflecting improved psychological well-being. The study concluded that music therapy could serve as a potential treatment for ADHD, offering benefits for both neurophysiological and psychological aspects.

[60] ([60]) explored the impact of active music therapy on adolescent boys with ADHD through an RCT. Participants were randomised into three groups: a waitlist control group or two active music therapy groups. The music therapy groups both involved instructional and improvisational music therapy. The music therapy interventions decreased levels of impulsivity, as evidenced by reduced errors in the Stroop Test (STT), indicating improved self-control and decision making. This supports the use of music therapy methods that involve active music-making as a beneficial intervention for managing ADHD symptoms.

Similarly, [61] ([61]) conducted an RCT to explore the impact of active music-making in music therapy on adolescent boys using the same three-group design as [60] ([60]). However, their sample specifically included aggressive adolescent boys, almost all of whom were diagnosed with ADHD. Although no statistically significant changes in aggressive behaviour were observed from the music therapy group, there were improvements in parent ratings of disruptive and antisocial behaviours. Although these improvements were not reflected in teacher assessments, they suggest the boys made improvements in social interactions, showing that music therapy might guide individuals to improve the way they relate to others and reduce aggressive behaviour.

Two studies explored the therapeutic potential of combining music therapy with CBT. [85] ([85]) conducted a study on adolescents with ADHD using an intervention that combined CBT with music-based emotion-regulation skills. The outcomes indicated that the intervention increased adaptive emotion-regulation strategies, such as cognitive reappraisal, and decreased maladaptive strategies, such as expressive suppression, suggesting it is an effective strategy for improving emotional well-being and coping mechanisms. Similarly, [89] ([89]) examined the effects of using a combined intervention of music therapy and CBT on a large sample of children with ADHD. The study was an RCT, so the children were randomised into the control and observation groups. The intervention group showed significant improvements in cognitive functions and attention, as well as reductions in behavioural problems and ADHD symptoms. Together, these studies highlight that combining music therapy with cognitive behavioural interventions can significantly reduce ADHD symptoms and enhance cognitive functions, offering a comprehensive approach to treatment.

In summary, music-based interventions and music therapy can benefit individuals with ADHD in a variety of forms. This review identified several effective approaches, including music and movement interventions, music and yoga interventions, music combined with CBT, and standalone music therapy. The studies reviewed reported improvements in inattention ([40]), hyperactivity/impulsivity ([42]; [60]; [85]; [89]), quality of life ([40]), and emotional regulation ([85]), along with reductions in behavioural problems ([60]; [89]) and symptoms of depression ([55]).

## 4. Discussion

### 4.1. Summary of Results

To our knowledge, this is the first systematic review and narrative synthesis to thoroughly examine the effects of ADHD on music comprehension and performance, as well as how various forms of music engagement, whether as a source of auditory stimulation or as a therapeutic intervention, affect individuals with ADHD. This review includes 20 studies identified through a comprehensive literature search, encompassing a total of 1170 participants, 830 of whom were diagnosed with ADHD or ADD. The studies include a diverse age range, from 2 to 56 years, and were published between 1981 and 2023, reflecting the evolving research landscape and growing understanding in this field. The studies were categorised into three broad groups: those investigating music performance and processing in this population, those exploring music as an additional source of stimulation for individuals with ADHD, and those assessing the therapeutic potential of music for ADHD.

#### 4.1.1. Music Processing and Performance

The studies reviewed suggest that individuals with ADHD exhibit distinct challenges in musical tasks, particularly those involving timing, rhythm, and the perception of complex auditory stimuli. Both [16] ([16]) and [59] ([59]) identify specific timing deficits in individuals with ADHD, such as difficulties in processing short time intervals and beat tracking, which are consistent with findings from other research highlighting temporal processing issues as a core characteristic of ADHD ([16]; [59]; [50]). These impairments in timing may contribute to broader cognitive challenges associated with ADHD, including difficulties in attention, impulse control, and executive function ([71]). However, despite these challenges, the reviewed studies highlight that deficits may not encompass musical improvisation and expression ([28]), which aligns with research suggesting that individuals with ADHD may exhibit creative strengths, particularly in unstructured and spontaneous tasks ([80]).

These findings help to explain the positive impact of music training, particularly rhythm training, on timing skills, impulsivity, and working memory for individuals with ADHD ([59]; [35]). Such interventions have shown improvements in abilities directly related to music, but also in other general cognitive abilities showing a transfer effect ([79]). Therefore, enhancing these skills through music-based interventions may offer a promising therapeutic approach.

Overall, these findings suggest a complex profile of deficits in music processing for individuals with ADHD, particularly in tasks involving timing, rhythm, and complex musical perception. This complexity underscores the potential of music-based interventions to ameliorate specific ADHD symptoms, offering a holistic approach to treatment that could improve cognitive, emotional, and behavioural outcomes ([84]).

#### 4.1.2. Music Listening

Overall, the majority of studies in this review suggest that listening to music can positively impact individuals with ADHD in areas including arithmetic performance, upright balance, reading comprehension, and behavioural outcomes, while also reducing motor activity and negative mood ([1]; [7]; [19]; [26]; [43]; [56]; [82]; [90]). Whilst it has been historically accepted that individuals with ADHD perform better in environments with reduced distractions ([2]), our review indicates that listening to music before or during a task can improve performance in a range of domains more effectively than silence.

Some of these findings align with the optimal stimulation theory, which posits that individuals with ADHD have lower baseline arousal and may benefit from increased environmental stimulation, such as music, to achieve an optimal arousal level and improve performance in monotonous tasks ([87]). For instance, studies by [1] ([1]) and [43] ([43]) demonstrated that music improved arithmetic and reading comprehension performance in ADHD groups while causing a decline in performance among TD controls, thus, supporting this theoretical framework. Conversely, studies by [7] ([7]) and [26] ([26]) reported that listening to music led to improvements in both ADHD and TD groups in upright balance and arithmetic performance, respectively, highlighting that further research is needed.

However, this review also revealed that listening to music may not always have a positive impact on individuals with ADHD, negatively affecting performance when tasks require significant cognitive resources and increasing activity levels in some environments. For instance, [20] ([20]) found that music listening negatively impacted the reading comprehension performance of children with ADHD during challenging tasks. This suggests that while arousal-increasing stimuli like music can be beneficial to a point, they may become counterproductive when overstimulation occurs ([26]). In contrast, [43] ([43]) emphasised the importance of the level of stimulation provided by music in determining its effects on individuals with ADHD. Their study found that calm music, with or without lyrics, improved reading comprehension and reduced HRV changes, indicating a calming effect on the autonomic nervous system. However, rhythmic music was less effective, suggesting that it may not provide the same cognitive or physiological benefits and could potentially lead to overstimulation. These findings align with the Optimal Stimulation Theory, which highlights the need for a balance in arousal levels ([86]). While music can help individuals with ADHD achieve this balance, overloading cognitive resources with challenging tasks or overly stimulating music may disrupt performance, limiting its benefits ([20]; [43]).

Some studies in college students (not ADHD patients) distinguished the effects of significant reduction in motor activities under similar conditions. However, [82] ([82]) suggested that the unexpected findings were a result of methodological issues, with the external factor of the duration of class sessions overriding the music intervention. A control group of hyperactive children who were not exposed to the independent variable would have been a way to control for this factor. However, both studies were limited by small sample sizes, highlighting the need for research to clarify whether and under what conditions music consistently reduces motor activity in individuals with ADHD.

#### 4.1.3. Therapeutic Effects of Music and Music Therapy

This review highlights that music and music therapy can offer significant benefits for individuals with ADHD in various formats, including music and movement interventions, combined yoga and music therapies, integrated cognitive–behavioural and music therapies, psychological therapies with music in the background and standalone music therapy. The studies reviewed reported notable improvements in core ADHD symptoms, such as inattention and hyperactivity/impulsivity, as well as enhancements in quality of life, emotional regulation, and reductions in behavioural problems and symptoms of depression ([40]; [42]; [55]; [60]; [61]; [85]; [89]).

The neurophysiological evidence presented by [40] ([40]), showing enhanced activity in specific brain regions, helps to explain the enhanced cognitive functions, attention, and emotional stability observed by suggesting that music therapy can stimulate both cortical and subcortical areas ([40]; [74]). Moreover, the review reveals that music interventions appear particularly effective for individuals with ADHD, as demonstrated by [89] ([89]), who found that children without ADHD did not experience similar improvements. This raises intriguing questions about the unique impact of music on ADHD-specific neurocognitive profiles ([89]).

However, not all studies reviewed found statistically significant outcomes. For instance, [61] ([61]) reported improvements in parent ratings of disruptive and antisocial behaviours that were not mirrored in teacher assessments, highlighting variability in perceived effectiveness across different settings ([61]). Additionally, there was considerable heterogeneity in the therapeutic interventions, with durations ranging from 8 to 16 weeks and differences in approach, including both active and passive methods. While variations in format and implementation have been thought to affect the comparability of results, [60] ([60]) found no significant difference between active and passive music engagement in music therapy sessions, suggesting that the type of musical experience may not be as critical as previously assumed ([60]; [46]).

Despite this heterogeneity, the reviewed studies are evidence of the potential value of the therapeutic use of music and music therapy as an adjunctive treatment for ADHD ([40]). Given its diverse positive outcomes and its ability to address both core and comorbid symptoms, such as anxiety and mood disorders, music therapy offers a multifaceted approach that aligns well with the complex, individualised needs of ADHD patients ([37]). Further research should focus on optimising intervention parameters, such as the frequency, duration, and type of music engagement (music therapy method), to maximise benefits and clarify the mechanisms through which music exerts its therapeutic effects.

### 4.2. Comparison with the Results of Previous Systematic Reviews

The findings from our review are consistent with those of the earlier systematic review by [45] ([45]), which focused specifically on music and ADHD, concluding that music generally has a positive impact on individuals with ADHD. Although there were differences in search strategies between our review and that of Martin-Moratinos et al., there was an overlap of 10 studies, leading to similar conclusions. Their review highlighted the benefits of music therapy, encompassing both active and passive engagement with music, while also acknowledging some differences in music performance and processing between individuals with ADHD and TD controls. Therefore, our review can be seen as an extension of the existing literature exploring the intersection of ADHD and music.

### 4.3. Limitations

A primary limitation of this review is the heterogeneity of ADHD diagnoses in the included studies. ADHD is a highly diverse neurological disorder, with considerable variation in symptoms even among individuals with the same diagnosis. Many studies lacked the specification of ADHD subtypes, which could have enabled a more homogeneous sample and clearer results for specific sub-groups. Moreover, studies such as [82] ([82]) and [19] ([19]) included participants who were not formally diagnosed with ADHD, further increasing sample heterogeneity. Windwer’s study used teacher ratings on the Conners Teacher Rating Scale (CTRS) for boys deemed hyperactive, at a time when the DSM-III had just been published, which introduced ‘Attention-Deficit Disorder’ (ADD) without encompassing the hyperactivity now recognised as a trait of ADHD ([82]). Similarly, [19] ([19]) investigated hyperactive boys diagnosed with hyperkinetic disorder or ADD, before these conditions were reclassified under ADHD in the DSM-III-R ([4]). Including such studies captures the evolving understanding of ADHD but complicates direct comparisons due to shifting diagnostic criteria ([57]). Moreover, this review pooled findings from studies on both ADD and ADHD ([9]; [19]) as ADD is typically classified under the predominantly inattentive presentation of ADHD ([9]). This variability in diagnostic criteria may undermine a precise understanding of the effects of music on ADHD, as populations studied across different periods and frameworks are not entirely comparable.

The presence of comorbidities among participants is another limitation. Studies included individuals with co-occurring conditions such as Developmental Coordination Disorder (DCD), Oppositional Defiant Disorder (ODD), Depression, and Obsessive–Compulsive Disorder (OCD) ([59]; [42]; [60]). Since 75% of individuals with ADHD have a co-occurring mental health disorder ([8]), including participants with comorbidities enhances generalizability to the broader ADHD population. However, this also dilutes the specificity, as the effects of music on ADHD could differ between those with and without comorbid conditions, complicating conclusions about music interventions’ efficacy for specific ADHD subgroups.

Medication use is another potential confounder. Medication is a common ADHD treatment that can influence both symptoms and responses to interventions such as music therapy ([8]). Some studies, like [19] ([19]), [55] ([55]), and [60] ([60]), included only medicated participants, which could skew results towards a uniform symptom presentation. Conversely, studies like [1] ([1]) and [56] ([56]) had mixed medication statuses, introducing variability in symptom profiles. Several studies did not specify medication use, adding uncertainty about whether observed effects were due to music interventions or medication ([9]). This variability presents challenges in isolating the specific effects of music therapy from those of pharmacological treatment.

The majority of studies examined in this review focused on children and adolescents, with limited data on adults. This gap is particularly evident in studies discussing the therapeutic effects of music and music therapy, where no adult studies were included, and in music listening studies, which only featured one involving adults ([9]). This lack of representation requires caution when generalising findings to adult populations, as the effects of music interventions may differ significantly across age groups. Future research should prioritise examining the effects of music in adults with ADHD to provide a more comprehensive understanding of its impact across the lifespan.

Small sample sizes also present a notable limitation in this review. Six out of the twenty included studies featured sample sizes under 20 participants, which warrants caution in interpreting these findings ([82]; [19]; [40]; [60]; [61]; [85]). Small samples are prone to greater variability and are less likely to detect significant effects, raising concerns about the generalizability and robustness of results. Larger, well-powered studies are essential to validate the effects of music on ADHD symptoms and ensure more reliable conclusions.

Gender differences can also complicate interpretation, as ADHD presents differently in males and females ([24]). Although five studies included only male participants, most employed mixed-gender samples without disaggregating results by gender. This lack of gender-specific analysis limits the ability to generalise findings and may obscure important differences in how music affects ADHD symptoms across sexes.

A limitation specific to the studies where music was an audio stimulation was methodological variations in how music was presented. About half of the studies presented music aloud through speakers, while others used personal earphones, some with noise-cancellation features. These differences could affect auditory perception and cognitive processing, influencing responses to music interventions ([53]). Such inconsistencies limit the comparability of outcomes across studies and complicate conclusions about music’s overall therapeutic efficacy for ADHD.

Additionally, there was considerable variability in the types of music used across these studies. Some studies examined different musical elements, such as tempo and lyrics ([43]; [20]), while others focused on a single music genre, like rock or rap ([1]; [19]). In contrast, other studies explored soothing nature sounds ([7]) or specific pieces of classical music, such as Mozart ([90]). This diversity complicates efforts to draw general conclusions about the effectiveness of music interventions for ADHD, as different music genres and sounds may uniquely affect attention, behaviour, and cognitive function.

Heterogeneity in methodologies among music therapy studies also presents a challenge. Variations in formats, such as music combined with movement, yoga, cognitive–behavioural therapy, or standalone music therapy, complicate attempts to pool results. Additionally, music therapy research is prone to bias due to the inability to blind participants and researchers to their assigned conditions. Many studies did not adhere to guidelines for reporting music-based interventions ([62]), failing to specify the music used or provide a rationale for its selection. These methodological inconsistencies limit the ability to draw comprehensive conclusions about the effects of music on individuals with ADHD.

Due to the heterogeneity of the study designs and missing information about the applied music, the study participants, the therapists, the researchers, conflicts of interest, and funding, we were not able to fully adhere to the PRISMA guidelines ([54]), which is reflected in the PRISMA checklist (see Appendix A). Specifically, we were not able to assess the certainty, the confidence, the risk of bias, the risk ratios, the mean differences, and the specific causes for heterogeneity for each outcome, and it was impossible to perform subgroup analyses, meta-regressions or sensitivity analyses to gauge the robustness of the synthesised results. Regarding the guidance for the narrative synthesis according to [58] ([58]), our review provides a preliminary synthesis of the included studies and explores relationships in the published data. However, due to the scarcity of clinical trials, we were not able to develop a theoretical model of how and why music-based interventions might work for people with ADHD, and we were not able to assess the robustness of the content synthesis.

Finally, this systematic review was not pre-registered.

### 4.4. Implications and Future Directions

This review supports the significant potential of music, both as auditory stimulation and as a therapeutic intervention, as a versatile and effective tool for alleviating ADHD symptoms and providing additional benefits ([43]).

Given the identified limitations, several recommendations for future research are proposed. First, there is a need for longitudinal studies with extended follow-up periods to evaluate the long-term effects of music therapy on ADHD, as the studies reviewed lacked such designs and follow-up assessments ([40]; [42]). It is crucial to determine whether the benefits of music therapy are sustained over time or are merely temporary ([42]).

Second, RCTs should be conducted to directly compare music therapy with standard treatments, such as CBT and ADHD medication. While some evidence suggests that combined CBT and music therapy can yield improvements across various domains ([85]; [89]), direct comparisons are needed to ascertain whether music therapy can be effective as a standalone treatment or if its efficacy is primarily as an adjunctive therapy. Studies should also evaluate the effects of music therapy independent of medication, given some studies included medicated participants ([55]; [60]).

Third, future RCTs should compare different music therapy methods, such as music and movement interventions, combined yoga and music therapy, and integrated cognitive–behavioural approaches. This would help to identify the most effective formats and provide a basis for developing evidence-based guidelines ([60]).

Fourth, research should further explore how variations in music intensity (e.g., volume, complexity) affect task performance. Although some studies have examined different musical elements ([43]; [20]), there remains a gap in understanding the specific impact of music intensity on tasks of varying difficulty. Future studies should aim to quantify the threshold at which music becomes detrimental to cognitive performance. The performance level might be influenced by the music genre, which should also be considered.

Fifth, to enhance the reproducibility and validity of findings, future research should standardise task types, difficulty levels, and music intensity. This would allow for more reliable comparisons across studies and more robust conclusions regarding the efficacy of music interventions.

Sixth, when conducting studies on the therapeutic use of music or music therapy, the terminology used should be as clear as possible. This can be achieved by adhering to music reporting guidelines ([62]; [54]) and by using specific terms such as “music therapy” by referring to national or international definitions. For example, according to the American Music Therapy Association (AMTA), music therapy is the clinical and evidence-based use of music interventions to accomplish individualised goals within a therapeutic relationship by a credentialed professional who has completed an approved music therapy programme ([3]). Manuscripts reporting the use of music therapy might refer to this definition by reporting the intervention, the therapeutic goals and the professional training of the music therapist.

Seventh, in the reported studies reporting the therapeutic use of music or music therapy, the documentation and report of side effects were neglected. However, music and music therapy can have side effects. For example, they can elicit negative emotions, traumatic memories or unpleasant thoughts ([32]). Therefore, future studies should treat pharmacological, psychosocial, artistic, and musical interventions equally by reporting side effects across all types of interventions.

Additionally, a consensus is needed on the biological underpinnings of ADHD and their measurement to better understand how music influences these factors. Future studies should investigate the impact of musical therapies on brain regions implicated in ADHD, which could inform the development of biologically informed treatment approaches ([52]; [73]).

Addressing these gaps in research will be crucial to establishing music therapy as a valid, evidence-based treatment for ADHD, either as a standalone intervention or in combination with other therapies. If these conclusions are validated, they could lead to significant practical applications, such as the development of a specialised music curriculum for individuals with ADHD to be implemented nationally ([42]).

## 5. Conclusions

This systematic review has assessed the current state of understanding of the relationship between ADHD and music, covering music processing, performance, listening, and therapy. The findings suggest individuals with ADHD possess specific deficits in timing and rhythm, suggesting that tailored music training may improve these deficits through the improvement of cognitive functions like working memory and impulsivity ([59]; [28]). Furthermore, music listening was shown to enhance attention, reading comprehension, and motor control while reducing negative mood, but may negatively impact performance on complex tasks, depending on the context ([1]; [20]). Finally, music therapy demonstrated improvements in ADHD symptoms, quality of life, and neurophysiological functioning, supporting its role as a therapeutic intervention ([40]; [42]). While these findings are promising, further research is needed in all three areas, specifically for music intervention research to standardise interventions, examine long-term effects, and directly compare music therapy with traditional treatments to better establish its efficacy in managing ADHD. These insights also highlight the potential for music to contribute to more holistic, non-pharmacological approaches, offering individuals with ADHD new avenues for enhancing cognitive functioning and overall well-being. Table 4 shows a condensed synopsis of the main findings of this systematic review.

## Figures and Tables

**Figure 1 behavsci-15-00065-f001:**
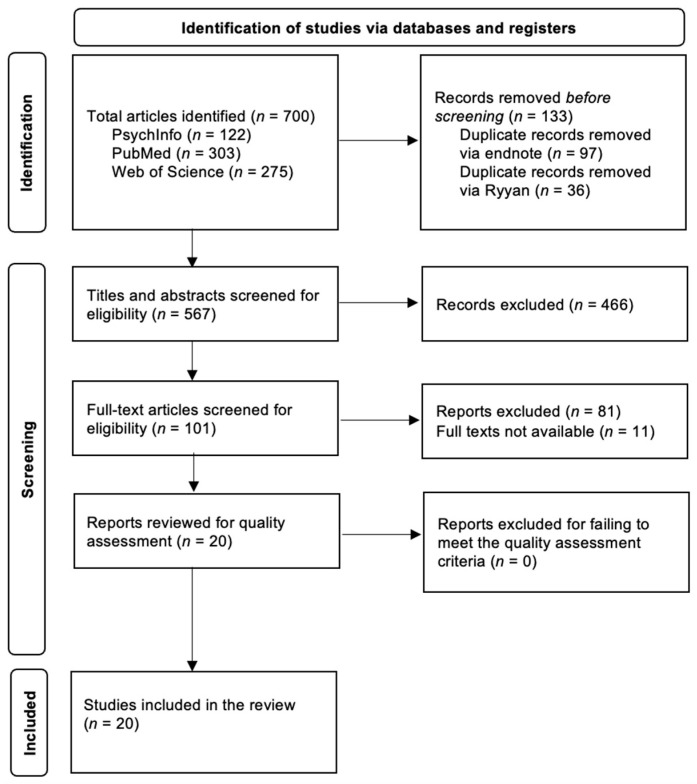
The PRISMA Flow Diagram of the selection process.

**Table 1 behavsci-15-00065-t001:** Music processing and performance: Detailed summary of included studies.

Author(s)/ (Year)/ Country	Sample/Group Size	Total Sample Size/Mean Age (Years)	Diagnosis and Medication	Outcome Measure(s)	Study Design	Main Outcomes and Statistical Significance	Overview
[16] ([16]) Brazil	The total sample (*n* = 36) was divided into three groups. Children with ADHD (*n* = 24) were subdivided into two groups; those not taking medication (*n* = 12), and those taking medication (*n* = 12). There was also a typically developing control group (*n* = 12).	*n* = 36 N/A (Range = 6–14 years)	ADHD was diagnosed according to the DSM-IV. 12/24 of the children with ADHD were taking medication (Methylphenidate).	(1) Spontaneous time- measured using the Spontaneous Time Test, (2) Time estimation with simple sounds- measured by the participants comparing the duration of 20 different sound-pairs, (3) Time estimation with music- measured by comparing the duration of 2 different songs.	Between subjects experimental design	The performance of both ADHD groups in time estimation of simple sounds in short time intervals was significantly lower than that of the control group (*p* < 0.05). In the task comparing musical excerpts of the same duration, the ADHD groups perceived tracks as longer when the musical notes had extended durations, in contrast to the TD controls. Despite these differences, all three groups demonstrated positive average performance across most tasks.	Children with ADHD exhibit deficits in time estimation and perception, especially in tasks involving musical rhythms.
[27] ([27]) Latvia (participants recruited from Germany)	The total sample of young adults (*n* = 75) was made up from musically naive participants (*n* = 25), music-educated participants (*n* = 25) and participants diagnosed with ADHD (*n* = 25) who were musically naïve.	*n* = 75 Mean age = 20.4 (SD = 1.7)	Diagnosis was not specified. Medication was not specified.	(1) Speech Perception- Measured by ability to discriminate between unfamiliar languages. (2) Short-Term Memory (STM)- measured using forward and backward digit span tasks. (3) Music perception ability- measured using the Advanced Measures of Music Audiation test.	Between subjects experimental design.	Individuals with ADHD exhibited significant deficits in complex music and language perception compared to controls (*p* < 0.001). No significant differences were found in simpler tasks or in STM capacity between the ADHD and control groups (*p* > 0.05). Despite their lower performance, individuals with ADHD overestimated their abilities, with a significant difference observed in self-assessment compared to controls (*p* < 0.001).	There are persistent challenges that individuals with ADHD face in complex auditory and linguistic tasks, underscoring the need for targeted interventions that simplify these tasks.
[28] ([28]) Latvia (participants recruited from Germany and Switzerland)	A total of 96 adolescents participated in this study; 19 adolescents with ADHD, 28 adolescents with ADD, 21 adolescents with dyslexia and 28 age matched unaffected controls.	*n* = 96 Mean age (ADHD) = 14.1 (SD = 1.4). Mean age (ADD) = 14.3 (SD = 1.8) Mean age (Controls) = 14.5 (SD = 1.1)	ADHD was diagnosed by a psychiatrist using the ICD-10-GM for ADHD and ADD. Medication was not specified.	(1) Music performance- measured by a music performance assessment scale (2) Neurophysiological correlates- assessed by magnetoencephalography (MEG).	Between subjects experimental design.	In terms of musical performance, adolescents with ADHD and ADD outperformed those with dyslexia in rhythmic reproduction, improvisation, and musical expression (*p* < 0.001). However, the control group surpassed ADD in rhythmic reproduction and ADHD in pitch and rhythmic improvisation (*p* < 0.05). Additionally, both the ADD and control groups performed better than ADHD in pitch reproduction (*p* < 0.05).	Adolescents with ADHD and ADD can perform on par with controls in rhythmic improvisation and musical expression suggesting that although ADHD may impair perception, it may not hinder creative processes.
[59] ([59]) France	Of the children recruited (*n* = 55), 41 were children with ADHD. Among them, 22 were children with ADHD only, and 19 were children with ADHD and DCD. A third group of TD children (*n* = 14) without ADHD were recruited for the control group. Of the adults recruited (*n* = 39), 21 formed the ADHD group and 18 participants formed the control group.	*n* = 94Children’s mean age = 8.9 (SD = 1.6) Adult’s mean age = 32.0 (SD = 9.0)	ADHD was diagnosed according to the DSM-5. No participants were treated with methylphenidate the day of the experiment.	Perceptual and sensorimotor timing skills- assessed with the Battery for the Assessment of Auditory and Sensorimotor Timing Abilities (BAASTA)	Between subjects experimental design	Children with ADHD performed worse in discriminating single durations compared to controls (*p* < 0.0001) and struggled significantly more than controls in detecting deviations from the beat in music (*p* < 0.05; *p* < 0.001). Both children and adults with ADHD had difficulties aligning to the beat in music (children: *p* < 0.0001; adults: *p* < 0.001). In motor tasks, children with ADHD, especially those with co-occurring DCD, showed poorer synchronisation (*p* < 0.00001).	Both children and adults with ADHD have core deficits in tracking the beat of music, consistent across different tasks. The variability in beat-tracking skills among ADHD participants suggests a heterogeneous cognitive impact.

**Table 2 behavsci-15-00065-t002:** Music listening: Detailed summary of included studies.

Author(s)/ Year + Country	Sample/Group Size + Mean Age (Years)	Diagnosis and Medication	Music Intervention	Type of Music	Experimental Conditions	Outcome Measure	Study Design	Main Outcomes and Statistical Significance	Overview
[1] ([1]) USA	Two groups of boys (*n* = 40) were made up of an sample with ADHD (*n* = 20) and grade matched TD controls (*n* = 20). Mean age = 9.9 years (Range = 7–13 years)	ADHD diagnoses according to the DSM-III-R. 6/20 of the children with ADHD were taking medication (Ritalin).	Listening to music as a form of high auditory stimulation. Compared with speech (low auditory stimulation) and silence (no auditory stimulation).	10 min videotape of subjects’ favourite music (98% was either rock and roll or rap music).	(1) 10 min of music, (2) 10 min of background speech, (3) 10 min of silence	Academic task performance- Mean arithmetic performance was the average of three scores; number of problems attempted, number of correct answers, accuracy.	Between subjects experimental design.	Children with ADHD scored significantly better in the music condition than in the speech (*p* = 0.005) or silence (*p* < 0.05) conditions. When the music condition was presented first, those with ADHD performed better. The control group had similar results under the 3 conditions.	The arithmetic performance of children with ADHD can benefit from music. Music can be seen to stimulate children with ADHD to do monotonous tasks.
[7] ([7]) Brazil	Outpatient clinic sample of children with ADHD (*n* = 26) and age and sex matched typically developing controls (*n* = 20). Mean age = 9.4 years (Range = 7–12 years)	ADHD diagnoses according to the DSM-IV-TR. No children with ADHD were taking medication.	Listening to music as a distractor. Compared with white noise as a distractor and silence as a non-distractor.	The music was composed of soothing ‘nature’ sounds.	(1) Silence, (2) relaxing music, (3) white noise.	Upright balance Performance- A Neurocom Balance Master Dynamic Posturography device was used to deliver a Sensory Organisation Test (SOT).	Between subjects experimental design.	Auditory distractors positively affected both groups. White noise was more effective than silence for children with ADHD (*p* = 0.001), especially in more challenging balance conditions. Relaxing music helped controls more than silence (*p* = 0.012).	Auditory distractors, such as music, may have enhancing effects on the upright balance performance of children with ADHD.
[19] ([19]) USA	The total sample (*n* = 8) was made up of males diagnosed with either Hyperkinetic Disorder or Attention Deficit Disorder (ADD). N/A (Range = 6–8 years)	All children were diagnosed with ‘Hyperkinetic Disorder’ or Attention Deficit Disorder. All 8 children were regularly taking prescribed stimulant medication (not specified).	Listening to music during two 20 min sessions; the first involved free play, the second involved structured activities.	The music consisted of Instrumental rock music.	(1) No music (2) Rock music	Child’s Behaviour, specifically their activity level and attention span- blinded observers recorded the behaviour of each child using a behaviour checklist.	Repeated measures factorial design	Statistically significant reduction (*p* = 0.005) in the number of motor activities during the music period. There were no significant differences regarding attention span.	Rock music can help to reduce motor activity in individuals with ADD.
[20] ([20]) China	The sample was randomly selected grade 1 students (*n* = 129) all with ADHD and from lower income families. Mean age = 6.2 years (SD = 0.5)	ADHD diagnoses according to the Chinese DSM-IV rating scale ([72]). Medication was not specified.	Listening to music with different musical factors (familiarity and tempo) as a form of auditory stimulation.	All music was Instrumental piano pieces of Chinese pop music.	(1) Easy reading without music, (2) easy reading with familiar and fast melody music, (3) easy reading with familiar and slow melody music, (4) easy reading with unfamiliar and faster melody music, (5) easy reading with unfamiliar and slow melody music, (6) difficult reading without music, (7) difficult reading with familiar an fast melody music, (8) difficult reading with familiar and slow melody music, (9) difficult reading with unfamiliar and faster melody music, (10) difficult reading with unfamiliar and slow melody music.	Poetry reading performance- The correct selection of a picture that completely matched the description of the poetry content.	Within subject experimental design	On the easy poetry reading task, whether background music was playing or not did not significantly enhance or inhibit students’ performance. On the difficult poetry reading task, students performed significantly better with no music than any music condition (*p* > 0.001). Also, in the music conditions, students generally performed better with unfamiliar music with a slower melody.	When a task does not cost a large amount of cognitive resources, listening to music can have an insignificant effect on reading comprehension. However, during a task that requires more cognitive resources, music can have a negative impact on reading comprehension. Although this impact is lessened with unfamiliar music with slower melodies.
[26] ([26]) South Africa	The sample (*n* = 42) was made up from children sampled from a remedial school with ADHD (*n* = 22) and children sampled from a public school without ADHD (*n* = 20). Mean age = 9.8 (Range = 8 to 10.9 years)	ADHD diagnoses according to a multi-disciplinary team. No children were taking medication for 24 h prior to the sessions.	Listening to music as an extra-task source of stimulation.	A 10 min CD was compiled for each child containing their favourite songs.	(1) Music, (2) silence.	Mathematical performance- measured by the number of problems correct, the number of problems attempted and an accuracy score.	Quasi-experimental comparison group design	A significant main effect was observed for the condition (music vs. silence) on accuracy (*p* < 0.006), with the music condition showing higher accuracy compared to the silence condition.	The presence of music can improve mathematical accuracy for both children with and without ADHD, supporting the optimal stimulation theory for all children.
[43] ([43]) Israel	A sample of preadolescents (*n* = 50); half with ADHD (*n* = 25) and half typically developing peers (*n* = 25). Mean age = 12.1 (SD = 1.2)	ADHD diagnoses established by a medical expert. Medication was not specified.	Listening to music as an extra-task source of stimulation.	The music included calm music without lyrics, calm music with lyrics and rhythmic music with lyrics.	(1) Without background music, (2) with calm music without lyrics, (3) with calm music with lyrics, (4) with rhythmic music with lyrics.	(1) Reading comprehension performance- tested via 5 multiple choice questions for each condition. (2) Heart Rate Variability (HRV)- using a portable monitor of HRV.	Between subjects’ experimental design	Reading comprehension significantly improved under the music condition in the ADHD group (*p* < 0.001) and deteriorated among the control group (*p* = 0.005). Heart rate variability in ADHD was significantly lower in the music condition (*p* < 0.05).	For children with ADHD, calm music can assist in regulating their autonomous responses and therefore enhance their reading comprehension performance. In TD children, listening to music can cause a distraction when reading and can debilitate their learning process.
[56] ([56]) USA	Expt 1- The sample of boys (*n* = 67) was made up from 41 boys with ADHD and 26 TD boys. Expt 2- The sample was made up of boys (*n* = 86) who all had ADHD. Expt 1- Mean age = 9.8 years (Range = 7.7–12.6 years) Expt 2- Mean age = 9.5 (SD = 1.4).	Expt 1- ADHD diagnosis according to the DSM-III-R. 36 of the boys with ADHD received two medication conditions; placebo and methylphenidate. Expt 2- ADHD diagnosis according to the DSM-III-R. 65 of the boys received two medication conditions; placebo and methylphenidate.	Expt 1 and 2- Listening to music as a distractor.	Expt 1 and 2- Each group voted for a contemporary music radio station (all groups chose radio stations that played rock or rap music).	Expt 1- (1) No distractor, (2) Music (3) Video. Expt 2- (1) No distractor, (2) Music.	Expt 1 and 2- Behavioural intervention performance- specifically looking at rule violations, seatwork completion and on task behaviour.	Expt 1- between subjects experimental design Expt 2- within subjects experimental design	Expt 1- Disruptive behaviour of children with ADHD was exacerbated by video condition but not in music condition. Boys in neither group (control and ADHD) were significantly distracted by music. In music condition, 61% with ADHD had no change, and 29% with had improved performance. Expt 2- In the music condition, 76% with ADHD had no change, and 15% with ADHD had improved performance.	Expt 1 and 2- Listening to music may help some children with ADHD more than a silent environment. Also, stimulant medication can significantly reduce distractor effects, bringing children with ADHD functioning to the level TD children.
[82] ([82]) USA	The sample consisted of male children (*n* = 13) identified and rated by teachers as hyperactive. N/A (Range = 5.5–8.5 years)	Hyperactivity was categorised by scores of 1.5 and above on the Conner’s Teacher Rating Scale (CTRS). Medication was not specified.	Listening to an ascending musical stimulus programme as background music.	A 7 min stereophonic ascending musical progression cycle.	N/A	Activity- measured by the Motor Activity Rating Scale (MARS).	Within subjects experimental design.	Results showed significant increase in activity from the baseline score to the ascending musical cycle score (*p* < 0.05) and again to the post-treatment score (*p* < 0.05).	The findings that hyperactive children show increased activity as a lesson progress may have been a function of the length of the class period rather than the presence of an ascending musical cycle.
[90] ([90]) Germany	A total of 84 participants were made up of 40 participants with ADHD and 44 healthy controls. Participants were randomly assigned to each experimental condition (music vs. silence). Mean age = 30 years	ADHD was diagnosed according to the DSM-IV and ICD-10 criteria. Patients with ADHD taking methylphenidate were told to stop the medication 48 h prior to the examination.	Listening to Mozart’s music for 10 min.	The piece of music used was the Mozart piano sonata for four hands (KV 440).	(1) Mozart’s music, (2) silence.	(1) Subjective arousal- Measured by the Global Mood-Arousal scale (2) Mood- Measured by the Current Mood Scale.	Randomised Control Trial.	Listening to Mozart’s music decreased negative mood in all groups (*p* > 0.05). In the ADHD group, a silent condition increased arousal (*p* = 0.046) and negative mood, (*p* = 0.004).	Music can significantly improve mood in individuals with ADHD. Conversely, silence can increase arousal and decrease positive mood in people with ADHD.

**Table 3 behavsci-15-00065-t003:** Therapeutic effect of music and music therapy: Detailed summary of included studies.

Author(s)/Year + Country	Sample/Group Size + Mean Age (Years)	Diagnosis and Medication	Music Intervention	Type of Music	Experimental Conditions	Outcome Measure(s)	Study Design	Main Outcomes and Statistical Significance	Overview
[40] ([40]) Taiwan	The study included 13 children with ADHD. Mean age = 6.09 (SD = 0.7)	ADHD was diagnosed in accordance with the DSM-V, and ADHD severity was evaluated using SNAP-IV. Medication was not specified.	The music and movement intervention lasted 1 h per week for 8 consecutive weeks. During the 1 h sessions, the participants learned to sing prominent Taiwanese nursery rhymes while concurrently performing corresponding body movements.	The music consisted of 6 nursery rhymes.	N/A	(1) Quality of life- via the Paediatric Quality of Life Inventory (PedsQL). (2) Core symptom severity- via the Conners Kiddie Continuous Performance (K-CPT 2) and the Swanson, Nolan, and Pelham Rating Scale (SNAP-IV). (3) Neuropsychological changes- determined with EEG recordings.	Within subjects pre-post intervention design	The participants’ quality of life increased significantly after the intervention (*p* < 0.05). Furthermore, the participants’ reaction times decreased after the intervention (*p* < 0.05). Finally, EEG analysis demonstrated an increase in alpha power and a decrease in delta power in some channels.	The music and movement intervention is an effective tool for ADHD treatment, significantly improving patients quality of life and attention.
[42] ([42]) China	Through a random sampling process, 60 kindergarten children with ADHD comorbid ODD were recruited and divided into 4 equal groups (*n* = 15). Mean age = 4.9 years (Range = 4–6 years)	Comorbid ADHD and ODD was diagnosed by a psychiatrist and confirmed by a school paediatrician. Participants did not take drugs for one day prior to the study.	The music intervention involved playing music that participants chose from some common children’s songs for 10 min in a quiet classroom. The yoga and music group received a 16-week yoga and music intervention (10 min of a yoga intervention and music intervention twice a week); the control group did not receive any intervention; the yoga-only group received a 16-week yoga intervention (10 min of a yoga intervention, twice a week); the music-only group received a 16-week music intervention (10 min of a music intervention, twice a week).	Participants chose their favourite track from some common children’s songs.	(1) 16-week yoga and music intervention, (2) No intervention, (3) 16-week yoga only intervention, (4) 16-week music only intervention.	Levels of inattention, hyperactivity/impulsivity and ODD- using the Chinese version of the MTA SNAP-IV ADHD rating scale, both parent and teacher-rated (pre-test and post-test).	Within subjects pre-post intervention design	The combined yoga and music intervention was the most effective intervention, significantly reducing inattention, hyperactivity/impulsivity, and ODD scores in the children (*p* < 0.05). The effectivity of the combined yoga and music intervention was followed by the yoga- and the music-only interventions, respectively.	The combined yoga and music intervention can help children with comorbid ADHD and ODD focus their attention and reduce hyperactivity/impulsivity and ODD behaviours.
[55] ([55]) Republic of Korea	A total of 36 participated in the experiment, consisting of an ADHD control group (*n* = 18) and ADHD music therapy group (*n* = 18). Mean age = 12.1 (SD = 2.5)	ADHD was diagnosed at a university hospital. Comorbid depression was diagnosed using the CDI Scale. All children continued to receive recommended drug therapy (not specified).	Music therapy was conducted for 3 months (24 50 min sessions in total) twice a week. The therapy included both active music therapy (improvisation) and receptive music therapy (music listening).	The music used for the music therapy consisted of motivating, relaxing, and both motivating and relaxing music and was selected through a preference survey.	(1) The control group received standard care in the form of drug therapy only, (2) the music therapy group received music therapy and standard care.	(1) Changes in serum markers- measured using blood samples. (2) Serotonin and cortisol levels- measured using high-pressure liquid chromatography (HPLC) and Cortisol Radioimmunoassay (RIA). (3) Heart Rate (HR)- measured using an automatic blood pressure monitor. (4) Levels of depression- using the Children’s depression inventory (CDI). (5) Daily stress- measured using the Daily hassles questionnaire (DHQ).	Randomised controlled trial (RCT)	The ADHD music therapy group’s 5-HT secretion increased (*p* < 0.001), whereas cortisol expression (*p* < 0.001), BP (*p* < 0.001) and HR (*p* < 0.001) decreased. The CDI and DHQ psychological scales also showed positive changes (*p* < 0.01 and *p* < 0.001, respectively). However, those with ADHD who did not receive music therapy showed no increase in 5-HT secretion, and cortisol expression, BP, and HR did not decrease. In addition, the CDI and DHQ psychological scales did not display positive changes.	The application of music therapy as an alternative treatment for ADHD can produce positive neurophysiological and psychological effects. The findings propose an alternative to medicine for preventing and treating comorbid depression through music therapy.
[60] ([60]) New Zealand	The sample was made up of 13 adolescent boys with ADHD. Students were randomised into three groups; a waitlist control group (*n* = 5), an improvisational music therapy followed by instructional music therapy group (*n* = 4) and an instructional music therapy followed by improvisational music therapy group(*n* = 4). Mean age = 13.0 (Range = 11–16)	ADHD was diagnosed according to the DSM-IV. All students were being treated with stimulant medication.	(1) Instructional Sessions: 8 structured sessions focused on teaching and modelling specific beat and rhythm tasks (2) Improvisation Sessions: 8 sessions where students freely improvised with various percussion instruments.	Music was created using percussion instruments only in both music therapy genres.	(1) The waitlist control group were not offered music therapy treatment, (2) A music therapy group had 8 sessions of improvisational music therapy followed by 8 sessions of instructional music therapy, (3) A music therapy group had the music therapies in reverse order.	(1) Impulsivity- measured using the Synchronised Tapping Task (STT). (2) Symptomatology improvement- measured using the Conner’s Global Index.	Randomised controlled trial (RCT).	The musical interventions significantly decreased STT errors when compared to the control group (*p* = 0.004). Furthermore, teachers reported a significant reduction in Conners’ DSM-IV Total and Global Index subscale scores for both interventions. There was no statistically significant difference between the instructional and improvisation treatments (*p* = 0.250).	Both instructional and improvisational music therapy may contribute to a reduction in a range of ADHD symptoms in the classroom, and increase accuracy on the STT showing a reduction in motor impulsivity.
[61] ([61]) New Zealand	The sample was made up of 15 aggressive adolescent boys. 12 boys had a diagnosis of ADD or ADHD. Adolescents were randomised into three groups: Music therapy group 1 (*n* = 6), Music therapy group 2 (*n* = 5) and waitlist control group (*n* = 4). Mean age 12.9 (Range = 11–13)	ADHD was diagnosed according to DSM-IV criteria. 9 boys were taking psychotropic medication.	The music therapy intervention was the same for group 1 and 2 and consisted of 16 sessions of 30–45 min, twice a week. Examples of activities included group music listening, singing personalised greeting songs, taking part in rhythm-based exercises, playing various instruments, and group songwriting.	N/A	(1) Music therapy group 1, (2) Music therapy group 2, (3) Waitlist control group.	Aggressive behaviour- measured using the Developmental Behaviour Checklist.	Randomised controlled trial (RCT).	Parent scores showed clear improvements across both ‘Disruptive’ and ‘Antisocial’ Developmental Behaviour Checklist subscales for both groups, whilst teacher results were less consistent, and the assessment of ‘Disruptive’ and ‘Antisocial’ data revealed no statistical differences. Within-session aggression was rarely observed, and subjects did develop positive relationships with peers.	Music therapy programmes may help some adolescents to interact more appropriately with others and assist in the development of positive relationships with peers.
[85] ([85]) Iran	The sample was made up of 8 adolescents with ADHD. Mean age = 15.87 years (Range = 13–17 years)	ADHD was diagnosed according to the DSM-5. Medication was not specified.	The intervention consisted of 12 weekly individual sessions, face-to-face, of standard CBT for adolescents with ADHD and adjunctive music-based emotion-regulation skills. The music-based intervention involved passive listening to pre-recorded music provided by clinicians.	Each participant was exposed to new-age genre music they had chosen for 15 min before the start of each CBT session, for 30 min during each CBT session as background music, and for 30 min each day as a between-session CBT homework.	N/A	(1) Levels of cognitive, emotional, and behavioural problems- measured using the Conners’ Parent Rating Scale. (2) Emotional regulation- measured using the Emotion-Regulation Questionnaire for Children and Adolescents (ERQ-CA)	Single-case experimental design.	The intervention was effective in reducing the core symptoms of ADHD, such that participants showed an increase in adaptive emotion-regulation strategies (cognitive reappraisal) (*p* < 0.05) and decrease in maladaptive emotion-regulation strategies (expressive suppression) (*p* < 0.05). The intervention was also found to be highly acceptable to participants.	The combination of standard CBT with music-based treatment designed to enhance emotion-regulation skills can augment the therapeutic benefits of CBT for adolescents with ADHD.
[89] ([89]) China	The sample was made up of 120 children with ADHD who were divided randomly into the control (*n* = 60) and observation (*n* = 60) groups. Mean age = 4.3 (Range = 2–7 years)	ADHD was diagnosed DSM-IV. Participants did not take drugs for one week prior to the study.	The observation group received music therapy combined with a cognitive behavioural intervention over a 16-week period. The intervention involved music therapists conducting weekly group music therapy sessions with five participants per group. The intervention focused on attention training, where patients concentrated on specific music cues while ignoring other stimuli. Activities included continuous attention exercises and both structured and improvised musical interactions.	N/A	(1) Control group, (2) Observational group.	(1) Improvement of symptoms—measured using the ADHD rating scale for parents (ADHD-RS-IV). (2) Attention functions—measured using the Numerical cross attention test (NCT). (3) Comparison ability—measured using the combined Raven’s test (CRT) (4) Cognitive flexibility—measured using the Wisconsin card sorting test (WCST). (5) Intelligence-measured using the Wechsler intelligence scale for children (C-WISC). (6) Behavioural problems—measured using Conner’s children behaviour scale for parents.	Randomised controlled trial (RCT).	After the intervention, the attention deficit, hyperactivity-impulsiveness, and ADHD-RS-IV total scores of the observation group were far lower than those of the control group (*p* > 0.05), showing a significant Improvement of symptoms. Moreover, the CRT, NCT, WCSC and C-WISC indices of the observation group significantly increased more than those of the control group (*p* < 0.05). In contrast, the relevant indexes of the control group did not show any significant changes after the intervention (*p* > 0.05).	Musicotherapy combined with a cognitive behavioural intervention can improve the cognitive functions of children with ADHD and therefore significantly improve symptoms highlighting its clinical application value.

**Table 4 behavsci-15-00065-t004:** Synopsis of the main findings of this systematic review on music and ADHD. * Mixed evidence. † Findings in adults.

Theme	Main Findings
Music performance and processing of people with ADHD	Difficulties in tasks involving timing, rhythm, and beat synchronisation † Performance difficulties in complex musical perception tasks † Strengths in musical improvisation and expression
Effects of listening to music in people with ADHD	Enhanced arithmetic performance Improved upright balance performance Enhanced reading comprehension performance * Reduced Motor Activity * Decreased negative mood †
Beneficial effects of music therapy in ADHD	Improved inattention and hyperactivity/impulsivity Improved quality of life Reduced behavioural problems Improved emotional regulation Prevention and treatment of depression

## Data Availability

Not applicable.

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
