# Peer review of "Exploring the Intersection of ADHD and Music: A Systematic Review"

_behavsci, 2025, doi:10.3390/bs15010065_

Round 1
Reviewer 1 Report
Comments and Suggestions for Authors
The authors have conducted a systematic review on a topic of great importance that has not yet been the subject of such detailed analysis. The manuscript is well structured and provides a clear description of the methodology.
I recommend that the manuscript be accepted for publication, although I have a few suggestions for improvement.
The majority of the studies examined children and young adults. It would be beneficial to reiterate this in the limitations section. Furthermore, categorising the main results in Table 4 by age group would enhance comprehension. Additionally, a sentence should be included in the discussion that contrasts the results in terms of children versus adults.
Author Response
Comment 1:
The authors have conducted a systematic review on a topic of great importance that has not yet been the subject of such detailed analysis. The manuscript is well structured and provides a clear description of the methodology. I recommend that the manuscript be accepted for publication, although I have a few suggestions for improvement. The majority of the studies examined children and young adults. It would be beneficial to reiterate this in the limitations section.
Response 1:
Thank you for your valuable feedback. We acknowledge the limitations associated with the fact that the majority of studies examined focus on children and adolescents, rather than adults. This is particularly evident in the sections discussing the therapeutic effects of music and music therapy, where no studies on adults were included, and in the music listening studies, which only featured one study involving adults. To address this, we have updated the limitations section to emphasise the caution needed when generalising results, especially in these two areas, to an adult population.
Comment 2:
Furthermore, categorising the main results in Table 4 by age group would enhance comprehension.
Response 2:
As Tables 1-3 already specify the age groups and the age range is reiterated within the body of the results, we did not initially feel it was necessary to categorise the findings in Table 4 by age. However, we have included the obelisk (†) symbol to highlight findings that were observed in adults, as these represent a minority of the studies. This symbol was chosen as the asterisk (*) has already been used to indicate mixed evidence.
Comment 3:
Additionally, a sentence should be included in the discussion that contrasts the results in terms of children versus adults.
Response 3:
The discussion now emphasises, in the limitations section, the need for further research on adults to address the current gap in studies focused on this demographic.
Reviewer 2 Report
Comments and Suggestions for Authors
This review aims to summarise studies investigating the relation between ADHD symptoms and music, both in terms of musical performance, music listening during cognitive performance, and as a therapy form. The research question does fill a gap in the literature and adds value in all three aspects of the use of music in ADHD. The paper is well structured and written, methodological procedures and concepts clearly explained and the limitations adressed.
The very few comments I have are in regards to the interpretation of the results from the second research question, i.e. listing to music while performing a cognitive task. Some studies in the general population (e.g. Souza & Barbosa, 2023) have distinguished effects of having background music with lyrics and without lyrics, and whether the lyrics have been in the listeners first language or not (Sun et al., 2024) which would be two variables impacting cognitive load. While lyrics was adressed in the study by Madjar (2020), included in this review, the results were not reported or commented on in the review. Perhaps this could be added somewhere in the discussion around lines 539-544, as it likely is a distinguishing factor interplaying with cognitive functioning in ADHD where music can increase arousal and motivation, but can also start becoming distracting if the music is demanding working memory resources (as described).
Souza, A. S., & Leal Barbosa, L. C. (2023). Should We Turn off the Music? Music with Lyrics Interferes with Cognitive Tasks. Journal of cognition, 6(1), 24. https://doi.org/10.5334/joc.273
Sun Y, Sun C, Li C, Shao X, Liu Q and
Liu H (2024) Impact of background music on
reading comprehension: influence of lyrics
language and study habits.
Front. Psychol. 15:1363562.
doi: 10.3389/fpsyg.2024.1363562
Another question that was clear to me was whether you tried to get in contact with the authors from the 11 articles that were excluded because of lack of access. Perhaps that could be added.
Another limitation that should be mentioned is that quite a few of the studies had a very small samples size (6 out of the 20 studies had sample sizes lower than 20 participants), warranting caution in its interpretations.
All in all, I think the review is an important contribution the field and may have implications that could benefit the individuals with ADHD and open up for more holistic approaches to improving cognitive functioning and life quality beyond medical treatment.
Author Response
Comment 1:
The very few comments I have are in regards to the interpretation of the results from the second research question, i.e. listing to music while performing a cognitive task. Some studies in the general population (e.g. Souza & Barbosa, 2023) have distinguished effects of having background music with lyrics and without lyrics, and whether the lyrics have been in the listeners first language or not (Sun et al., 2024) which would be two variables impacting cognitive load. While lyrics was adressed in the study by Madjar (2020), included in this review, the results were not reported or commented on in the review. Perhaps this could be added somewhere in the discussion around lines 539-544, as it likely is a distinguishing factor interplaying with cognitive functioning in ADHD where music can increase arousal and motivation, but can also start becoming distracting if the music is demanding working memory resources (as described).
Souza, A. S., & Leal Barbosa, L. C. (2023). Should We Turn off the Music? Music with Lyrics Interferes with Cognitive Tasks. Journal of cognition, 6(1), 24. https://doi.org/10.5334/joc.273
Sun Y, Sun C, Li C, Shao X, Liu Q and Liu H (2024) Impact of background music on reading comprehension: influence of lyrics language and study habits. Front. Psychol. 15:1363562. doi: 10.3389/fpsyg.2024.1363562
Response 1:
Thank you for your insightful comments on cognitive load. We have inserted the two references by writing "Some studies in college students (not ADHD patients) distinguished the effects of background music with lyrics and without lyrics, and with lyrics in different languages [80, 81]. Interestingly, Souza and Barbosa [80] found that instrumental music alone neither hindered nor improved memory, reading comprehension, and arithmetic exercises. However, lyrics had an impedimental effect [80]. Similarly, Sun et al. [81] reported findings indicating that listening to pop music with lyrics reduces reading comprehension. Additionally, they found that reading comprehension was reduced more by lyrics in the same language as the written texts [81]. However, it should be considered that these two studies were done in college students, not in people with ADHD."
However, Madjar's (2020) findings suggest that in individuals with ADHD, the type of music that influences cognitive load is more related to the calm/rhythmic nature of the music rather than the presence or absence of lyrics. We have added this nuance to the results and discussion section, where I outline the influence of music type on cognitive load in ADHD, which we agree adds valuable context. Additionally, we have expanded on how the impact of music type on cognitive load complements the findings from Dong's study, further emphasising the complexity of music’s role in cognitive tasks in ADHD. We also mention that research should further explore how variations in music intensity (e.g. volume, complexity) affect task performance, which links to your comments regarding lyrics and language.
Comment 2:
Another question that was clear to me was whether you tried to get in contact with the authors from the 11 articles that were excluded because of lack of access. Perhaps that could be added.
Response 2:
We have expanded the methodology section to clarify the lack of access protocol used in the review: “Unfortunately, the full text of 11 studies could not be accessed, despite extensive efforts to obtain them, including contacting authors when possible and searching multiple databases. As a result, these studies were excluded from the final analysis.”
Comment 3:
Another limitation that should be mentioned is that quite a few of the studies had a very small samples size (6 out of the 20 studies had sample sizes lower than 20 participants), warranting caution in its interpretations.
Response 3:
We have updated the limitations section to include a note regarding the sample size of 6 out of 20 studies being under 20 participants, which warrants caution in the interpretation of those findings.